# Approaches and results of intersectoral actions for tuberculosis control in the world: A scoping review

Rosiane Davina da Silva  [1]*, Erica Rayane Galvão de Farias[2],
José Mateus Bezerra da Graça[1], Eslia Maria Nunes Pinheiro[1],
Elisângela Franco de Oliveira Cavalcante[1]

**1** Department of Public Health, Graduate Program in Public Health, Federal University of Rio Grande do Norte, Natal, Rio Grande do Norte, Brazil, **2** Nursing Department, Graduate Program in Health and Society, Federal University of Rio Grande do Norte, Natal, Rio Grande do Norte, Brazil

\* rosiane.uepb@outlook.com

## Abstract

### Background

Tuberculosis is a neglected disease with a wide global scope that overcomes public health challenges, also constituting an obstacle to social development. In the effort to control the disease, Tuberculosis Control Programs around the world have aligned their actions with the World Health Organization End TB Strategy, which emphasizes intersectorality as a fundamental component for effective disease control.

### Objective

To map the approaches and results of intersectoral tuberculosis control actions at the global scenario.

### Methodology

This scoping review followed the PRISMA (Preferred Reporting Items for Systematic Reviews and Meta-Analyses) guidelines and the Joanna Briggs Institute manual, ensuring methodological rigor and transparency. The review protocol was registered in the Open Science Framework. Searches were carried out in indexed databases and in the gray literature. Data collection took place by two independent reviewers, with results stored and organized in spreadsheets.

### Results

Three hundred and ninety-six (396) studies were identified, of which 60 were analyzed in full, resulting in the inclusion of 11 studies for the final review. It was evidenced that intersectoral articulation is fundamental in tuberculosis control, involving sectors such as health, education, social assistance and justice, to ensure adequate

**Data availability statement:** All relevant data are within the manuscript and its Supporting Information files.

**Funding:** This study was financed in part by the Coordenação de Aperfeiçoamento de Pessoal de Nível Superior - Brasil (CAPES) - Finance Code 001. The funders had no role in study design, data collection and analysis, decision to publish, or preparation of the manuscript.

**Competing interests:** The authors have declared that no competing interests exist.

health care and social support, particularly for vulnerable populations. Community education and awareness played a central role in treatment adherence and reducing stigma, while resource mobilization was needed to maintain health services, especially in contexts of scarcity.

## Conclusion

The integration of intersectoral services, involvement of non-governmental organizations and active community participation are essential elements for effective tuberculosis control. The findings reinforce the importance of addressing the social determinants of health to achieve the objectives of the End TB strategy, promoting an environment conducive to the prevention, early detection and effective treatment of the disease.

## Introduction

Tuberculosis is a neglected infectious disease that affects millions of individuals annually in the world. Its incidence is closely related to the social determinants of health, which manifest themselves in cycles of poverty and social inequalities. Such factors increase the risk of disease spread in the less favored population, exacerbating socioeconomic difficulties and health inequities [1].

Thus, the confrontation of tuberculosis on the global scenario is not only a public health problem, but also a social development challenge [2]. In this perspective, the End TB Strategy post-2015 was approved, which establishes as a goal "a world free of tuberculosis: zero death, illness and suffering related to tuberculosis". Therefore, the goal is to end the global epidemic of the disease by 2035, in line with sustainable development goal that aims to control the tuberculosis epidemic by 2030 [2,3].

The End TB Strategy requires the union of countries and national tuberculosis control programs. This involves efforts to confront the disease through universal access to prevention and individual-centered care. The strategy highlights the importance of innovation and intersectoral action, requiring the involvement of national leaders and co-participation in intersectoral actions to address the social determinants of health [1,4].

Intersectorality can be understood as the articulation and integration between different sectors, its knowledge and powers, aiming to face complex problems, in order to overcome the fragmentation of knowledge and social structures to generate significant impacts on the health of the population. In the health field, more than a concept, it is a social practice that arises from dissatisfaction with the health sector's responses to contemporary challenges that require the articulation of other sectors [5].

Thus, intersectorality emerges as an important mechanism of democratic public management, as it seeks the co-participation and articulation of intersectoral actions in political decisions and in the complementary of actions through diagnosis, programs, actions and shared responsibilities. Thus, by contemplating the needs of the individuals who demands public assistance in their entirety, intersectorality can provide substantial improvements in the living conditions of the population by combating

social inequalities and health inequalities. However, it is still characterized as an important challenge for the various levels of management [6–8].

In this perspective, this study aims to map the approaches and results of intersectoral tuberculosis control actions at the global scenario. To this end, we mapped and summarized the approaches and results of intersectoral actions in tuberculosis control, with the aim of identifying the best practices that can guide the development and implementation of more effective interventions to combat the disease.

## Materials and methods

This is a scoping review, built according to the methodological trajectory developed by the Joanna Briggs Institute Reviewer's Manual for Scoping Reviews and following the recommendations of the Preferred Reporting Items for Systematic Reviews and Meta-Analyses extension for Scoping Reviews Checklist (PRISMA-ScR) [9,10].

This review was registered on the Open Science Framework (OSF) platform on February 27, 2024, under the email address: https://osf.io/n2gxu/. It introduces an abstract containing title, authors, objective, research question, search strategy in MEDLINE, the consulted databases and the collected variables that can be accessed directly at the provided link. OSF is free software accessible to the academic community, which aims to promote open collaboration in scientific research, supporting researchers to work on their projects privately or publicly, with the objective of expanding the access, integrity and reproducibility of academic research from planning to dissemination [11].

The formulation of the research question followed the mnemonic combination: Population, Concept and Context (PCC), as suggested by the method of the Joanna Briggs Institute Reviewer's Manual. The research question defined was: What is the scientific evidence on the approaches and results of tuberculosis control actions on the global scenario, focusing on intersectorality?

• Population (P): Tuberculosis

• Concept (C): Intersectorality

• Context (C): National Tuberculosis Control Programs, National Tuberculosis Control Policy and Tuberculosis Control Actions, Public Health.

The electronic capture of studies occurred in the MEDLINE (Medical Literature Analysis and Retrieval System Online), CINAHL (Cumulative Index to Nursing and Allied Health Literature), Web of Science, Embase (Excerpta Medica dataBASE) and Scopus databases. Access to these databases occurred through the Portal of Journals of the Coordination for the Improvement of Higher Education Personnel (CAPES), through the access of the Federated Academic Community (CAFe) with the login of the Federal University of Rio Grande do Norte, at the electronic address: https://www-periodicos-capes-gov-br.ezl.periodicos.capes.gov.br/.

The Latin American and Caribbean Literature on Health Sciences (LILACS) was also included, which was accessed through the Virtual Health Library, at the following electronic address: https://lilacs.bvsalud.org.

The search in the gray literature was conducted in the following portals:

• Brazilian Digital Library of Theses and Dissertations https://bdtd.ibict.br/vufind/

• Portugal Open Access Science Repository (RCAAP): https://www.rcaap.pt/

• WHO Global Research: https://www.who.int/publications/i

In order to perform this review, the Dart-Europe <http://www.dart-europe.eu/basic-search.php> and the Electronic Theses Online Service (EThOS)<https://ethos.bl.uk/Home.do;jsessionid=D21ECD249926BDE2B0B862D7670ED2E7> were considered; however, due to technical problems, they are disabled for data collection, from the beginning of the research until 28/06/2024.

Data collection took place in three phases:

First phase: The descriptors that composed the PCC mnemonic were selected from descriptors of Medical Subject Heading (MeSH), health sciences descriptors (DeCS) and Embase subject headings(EMTREE). Uncontrolled terms and singular and plural keywords were also included. The construction of the descriptors and search strategies was supported by a librarian.

Second phase: The search strategies in the databases are applied, performing tests and analysis of the titles and abstracts of the selected studies, as well as the terms of the index used to describe the articles. This process took place from February 10 to March 27, 2024.

Third phase: A new search was carried out with the identified keywords and the index terms in all the selected databases to extract the studies for analysis, finalized on March 15, resulting in 396 studies for analysis. More information can be accessed in the supporting information **S1 Support Files –** S1 Search Strategies. The document gives details of the process of electronically capturing studies in the databases.

Table 1 presents the descriptors used in the elaboration of the search strategies, which was initially developed in MEDLINE and later adapted to meet the specificities of each database.

The reference lists of all selected studies were reviewed for further analysis. To maintain consistency in the search for studies and avoid selection biases, the descriptors were used separately and associated, respecting the specific characteristics of each selected database.

Original articles, reviews, theses, dissertations, epidemiological bulletins, manuals, technical and governmental publications made available in full in English, Portuguese, Italian, Spanish and French were included. Duplicate articles, editorials, experience reports, annals, theoretical essays, reflection studies and books were excluded, as well as studies that did not meet the problem of this research. The time limit was defined from 2015 to 2024, considering the publication of the "End TB Strategy" by the WHO in 2015, which motivated the reorganization of tuberculosis control programs.

After searching the literature, all publications identified in the databases were exported in RIS format to Rayyan QCRI, through the following electronic address https://rayyan.ai/users/sign_in, where duplicates were deleted. The remaining publications were submitted to a selection process by two independent reviewers who read the titles and abstracts. Disagreements and doubts in the selection were resolved by an additional reviewer and by reading the materials in full. The process of selection and inclusion of the studies was presented in a flowchart, as proposed by PRISMA- ScR 2020.

Data extraction was performed manually from the selected studies using a structured form in Microsoft Excel 2016 spreadsheets containing the following variables: authors, journal name, and year of publication, country of study, type of publication, study design, objective, study population/sample and main results about the approaches results of intersectoral actions in tuberculosis control by two independent researchers as included in the supporting information S2 File**: Spreadsheets for storing data**. The results were submitted to a narrative synthesis and presented graphically to allow the visualization of the most discussed repercussions on the use of intersectorality in tuberculosis control.

## Results

A total of 396 studies were identified through the search strategies implemented, of which 60 were submitted to full evaluation. Among these, 11 studies were selected because they met the inclusion and exclusion criteria established for the final review. The main reason for the exclusion of the other studies was non-compliance with the objective of the study, as detailed in S3 Fig. **Checklist Prisma-ScR.** of the study selection process.

Table 2 presents a summary of the authors, year of publication, type of publication, study approach, database, journal, country, study objective and population analyzed in different studies related to tuberculosis control. It was noted that most of the studies found were research articles published in the MEDLINE and Scopus databases.

Table 3 summarizes the approaches and results of intersectoral actions aimed at tuberculosis control. The studies showed that collaboration between various sectors, such as health, education, social assistance, work and justice, as well

**Table 1. Strategies used in the search for publications to carry out the scoping review on the approaches and results of tuberculosis intersectoral control actions, on the global scenario, in the MEDLINE database. Natal, RN, Brazil, 2024.**

| Search strategies (MEDLINE) | |
|---|---|
| **#1 (P)** | ("tuberculosis"[MeSH Terms] OR "tuberculosis"[All Fields]) OR ("tuberculosis"[MeSH Terms] OR "tuberculosis"[All Fields] OR "tuberculoses"[All Fields]) OR "Kochs Disease"[All Fields] OR "Koch's Disease"[All Fields] OR "Koch Disease"[All Fields] OR "Mycobacterium tuberculosis Infection"[All Fields] OR "Infection, Mycobacterium tuberculosis"[All Fields] OR "Infections, Mycobacterium tuberculosis"[All Fields] OR "Mycobacterium tuberculosis Infections"[All Fields] |
| **#2 (C)** | "Intersectoral Collaboration"[All Fields] OR "Intersectoral Collaborations"[All Fields] OR "Intersectoral Cooperation"[All Fields] OR "Cooperation, Intersectoral"[All Fields] |
| **#3 (C)** | "Comprehensive Health Care"[All Fields] OR "National Health Programs"[All Fields] OR "Population Health Management"[All Fields] OR "Health Care, Comprehensive"[All Fields] OR "Comprehensive Healthcare"[All Fields] OR "Healthcare, Comprehensive"[All Fields] OR "Health Program, National"[All Fields] OR "Health Programs, National"[All Fields] OR "National Health Program"[All Fields] OR "Program, National Health"[All Fields] OR "Programs, National Health"[All Fields] OR "Health Services, National"[All Fields] OR "Health Service, National"[All Fields] OR "National Health Service"[All Fields] OR "Service, National Health"[All Fields] OR "Services, National Health"[All Fields] OR "National Health Services"[All Fields] OR "Public Health"[All Fields] OR "Community Health"[All Fields] OR "Health Management, Population"[All Fields] OR "Management, Population Health"[All Fields] |
| **#1 #2 #3** | (((("tuberculosis"[MeSH Terms] OR "tuberculosis"[All Fields]) OR ("tuberculosis"[MeSH Terms] OR "tuberculosis"[All Fields] OR "tuberculoses"[All Fields]) OR "Kochs Disease"[All Fields] OR "Koch's Disease"[All Fields] OR "Koch Disease"[All Fields] OR "Mycobacterium tuberculosis Infection"[All Fields] OR "Infection, Mycobacterium tuberculosis"[All Fields] OR "Infections, Mycobacterium tuberculosis"[All Fields] OR "Mycobacterium tuberculosis Infections"[All Fields]) AND ("Intersectoral Collaboration"[All Fields] OR "Intersectoral Collaborations"[All Fields] OR "Intersectoral Cooperation"[All Fields] OR "Cooperation, Intersectoral"[All Fields])) AND ("Comprehensive Health Care"[All Fields] OR "National Health Programs"[All Fields] OR "Population Health Management"[All Fields] OR "Health Care, Comprehensive"[All Fields] OR "Comprehensive Healthcare"[All Fields] OR "Healthcare, Comprehensive"[All Fields] OR "Health Program, National"[All Fields] OR "Health Programs, National"[All Fields] OR "National Health Program"[All Fields] OR "Program, National Health"[All Fields] OR "Programs, National Health"[All Fields] OR "Health Services, National"[All Fields] OR "Health Service, National"[All Fields] OR "National Health Service"[All Fields] OR "Service, National Health"[All Fields] OR "Services, National Health"[All Fields] OR "National Health Services"[All Fields] OR "Public Health"[All Fields] OR "Community Health"[All Fields] OR "Health Management, Population"[All Fields] OR "Management, Population Health"[All Fields]) |

Source: Research Data, Natal-RN, 2024.

as the performance of non-governmental institutions, community support and partnerships between the public and private sectors, resulted in significant improvements in the detection of the disease and adherence to treatment, especially in vulnerable populations.

The following figure represents a summary of the findings presented in Table 3, included in supporting file S4 Fig. **Conceptual structure of approaches and results of intersectoral actions in Tuberculosis control, Natal RN, 2024**. The figure presents the visual synthesis of the approaches and results identified in the scoping review. It structures and systematizes the information to improve the understanding of the mapped intersectoral approaches and results.

## Discussion

The analysis of the reviewed studies shows that intersectoral articulation has played a relevant role in tuberculosis control, especially in vulnerable populations, such as people deprived of liberty, co-infected with HIV, migrants and truck drivers [12,13,16,19,21,22]. These interventions, which involve collaboration between the health, justice and social assistance sectors, have enabled the implementation of individualized actions, taking into account the multiple dimensions that impact the health and well-being of these groups.

The effectiveness of these initiatives is reflected in the positive results achieved, since tuberculosis control requires a holistic approach, which covers not only clinical treatment, but also the social and economic conditions of the affected individuals [23]. In this sense, it is necessary to understand health as a social phenomenon, which implies that health professionals must establish solid links with people with tuberculosis and their families, aiming to overcome barriers to treatment adherence and improve care strategies [24].

**Table 2. Summary of authors, year of publication, and type of publication, study approach, database, journal, country, study objective and population.**

| N | Authors (Year) | Type of publication/Study approach (database) | Journal (Country) | Objective | Sample |
|---|---|---|---|---|---|
| 1 | Vries et al, 2017 [12] | Research Article Qualitative (MEDLINE) | BMC Public Health (4 European Union countries: Austria, Bulgaria, Spain and the United Kingdom) | To identify the health system factors that influence the treatment outcomes of people with multidrug-resistant tuberculosis in four European Union countries. | 35 and included policy and planning authorities, health care providers, and civil society organizations involved. |
| 2 | Nhassengo et al, 2023 [13] | Research/ Qualitative Article (MEDLINE) | BMJ Open (Republic of Mozambique) | To understand the perspectives of policymakers in the health and social support sectors on possible solutions to mitigate the financial impact among people with TB and their households in Mozambique. | 27 Health Policy Makers. |
| 3 | Boulanger et al, 2016 [14] | Research Article/ Literature Review (MEDLINE) | Clinical Infectious Diseases (Canada) | To provide an overview of the most important ethical considerations related to TB prevention, including emerging considerations, from the point of view of health workers in resource-limited settings. | It does not reveal how many studies were included in the review. |
| 4 | Weil et al, 2018 [15] | Journal Article/ Literature review (MEDLINE) | BMC Medicine (Switzerland) | To analize precedents for leadership in action against tuberculosis and highlights opportunities for bolder accountability and collaboration, especially at the national level, to drive action and impact. | It does not reveal how many studies were included in the review. |
| 5 | World Health Organization, 2022 [16] | Technical/ Mixed Manual (WHO Global Research) | World Health Organization (Switzerland) | To report on the adaptation and implementation of the WHO multisectoral accountability framework to end tuberculosis, highlighting best practices in this context. | It does not reveal how many countries were included in the analysis. |
| 6 | Rahevar et al, 2021 [17] | Research paper/ Qualitative/ (WHO Global Research) | Western Pacific Surveillance and Response (Eastern Pacific Region. Cases from China, Japan, Mongolia and the Republic of Korea) | To report a series of experiences in response to outbreaks of tuberculosis in schools in the Eastern Pacific region based on four case studies compiled by WHO collaborating centers (in China, Japan, and the Republic of Korea) and the Mongolian Ministry of Health. | Students and school staff, specifically in contexts of tuberculosis outbreaks (It does not mention quantity). |
| 7 | World Health Organization, 2015 [18] | Technical/ Mixed Report (WHO Global Research) | World Health Organization/ Global (Case studies in countries such as Myanmar, Philippines, Turkiye, Vietnam, Pakistan, India, Nigeria, Azerbaijan, among others) | To share experiences and approaches implemented in different countries for the involvement of health care providers in the management of drug-resistant tuberculosis, promoting a public-private combination. | It does not apply specifically to a defined study population, as the document compiles experiences from several countries. |
| 8 | Spruijt et al, 2019 [19] | Original/ Mixed Article (SCOPUS) | European Respiratory Journal (The Netherlands) | To evaluate the implementation of a screening and treatment program for latent tuberculosis infection (LTBI) among asylum seekers in the Netherlands. | 719 asylum seekers aged 12 and over living in asylum seeker centre from countries with an incidence of TB of more than 200 per 100,000 inhabitants. (Participated in the quantitative analysis) Of these 21 were interviewed. |

*(Continued)*

**Table 2.** (Continued)

| N | Authors (Year) | Type of publication/Study approach (database) | Journal (Country) | Objective | Sample |
|---|---|---|---|---|---|
| 9 | Yuen et al, 2019 [20] | Original/ Quantitative Article (SCOPUS) | PloS ONE (Peru) | To determine whether a community-based monitoring intervention could improve the management of contacts with tuberculosis by increasing adherence to and completion of preventive therapy. | 314 household contacts and 109 index cases with tuberculosis who started treatment between September 2015 and June 2016 in Lima, Peru. |
| 10 | Pescarini et al, 2017 [21] | Commentary/ Qualitative Article (SCOPUS) | Globalization and Health volume (Brazil and the United Kingdom) | To discuss some aspects of tuberculosis and migration control in the context of middle-income countries, as well as the possibility of implementing equitable and comprehensive policies in response to these issues. | 36 studies based on literature review and policy analysis were included for the final review. |
| 11 | Sharma et al, 2019 [22] | Original/ Quantitative Article (SCOPUS) | Indian Journal of Tuberculosis (India) | To evaluate the feasibility and outcomes of integrating TB screening and treatment services for truck drivers using the existing infrastructure of STI (Sexually Transmitted Infections) clinics and integrated counseling and testing centers (ICTC). | The study population consisted of all truck drivers and other associated persons at Sanjay Gandhi Transport Nagar in Delhi, India. It included a total of 14,644 truck drivers and 1,444 other people. Among these, 297 truck drivers and 30 additional individuals were referred for tuberculosis testing. |

Source: Research Data, Natal-RN, 2024.

The provision of personalized care, adapted to the specific conditions of different population groups, has been essential to expand access to diagnosis and treatment of tuberculosis. In this context, interventions that use technologies, such as text messages and alarms, were effective in intensifying promoting adherence to treatment among migrants, as highlighted by Alipanah et al. [25].

These initiatives reinforce the importance of the principle of health equity, which seeks to ensure that all people, regardless of their social, economic or working conditions, can access health services fairly and adequately. Equity refers not only to the equal provision of services, but to the adaptation of care strategies to meet the specific needs of each group, ensuring that individual or structural barriers do not prevent access to necessary care [26].

This is particularly relevant in tuberculosis control, as marginalized groups or those with atypical living characteristics, such as migrants and truck drivers, often face additional challenges that require tailored solutions to ensure adherence and treatment success [12,21,22]. Regarding truck drivers, the provision of health services in the workplace was an effective strategy to facilitate the diagnosis and treatment of tuberculosis [22].

The articulation between the health and justice, represented by prisons, sectors proved to be essential for the control of tuberculosis in prisons, especially in cases of multidrug-resistant tuberculosis (MDR-TB). In countries such as Austria, Spain and the United Kingdom, the screening and referral of people deprived of liberty with MDR-TB to health services have been shown to be key to increasing detection and ensuring adequate treatment of this population [12]. In Brazil, the introduction of molecular diagnostics in prisons has also contributed to early detection and effective treatment. Increasing incarceration rates intensify the impact of tuberculosis on these populations, pushing the boundaries of prisons and affecting the community [16].

While these actions are important, they need to be confronted with the global World Health Organization's "End TB Strategy", which aims to control the tuberculosis epidemic as a public health problem by 2035 through a more

**Table 3. Mapping of approaches and results of intersectoral actions for tuberculosis control on the global scenario. Natal-RN, 2024.**

| Study | Intersectoral action approaches for tuberculosis control | Results of intersectoral actions for tuberculosis control |
|---|---|---|
| Vries et al, 2017 [12] | • Care for vulnerable groups: Prisoners and Migrants with individualized actions;<br>• Screening and Referral of people with multidrug-resistant tuberculosis from the Prison Sector to health clinics in Austria, Spain and the United Kingdom;<br>• In Bulgaria, existence of a health clinic within prisons to assist people with tuberculosis;<br>• Community-Based Initiatives: to raise awareness about tuberculosis, identify cases and provide treatment support, including hard-to-reach groups (such as people in social vulnerability, migrants, homeless people, deprived of liberty, drug users and other marginalized populations);<br>• Involvement of Non-Governmental Organizations (NGOs) in the provision of Care: in Spain (Directly Observed Treatment, Assistance in obtaining housing and employment, as well as support for migrants). | • Approaches related to health actions in the prison sector and community-based initiatives were significant in expanding case detection, therapeutic care and achieving better health outcomes;<br>• NGOs played an important role in providing support and therapeutic adherence to people with multidrug-resistant tuberculosis. |
| Nhassengo et al, 2023 [13] | • Care for vulnerable groups: for people at economic risk and with HIV with individualized actions;<br>• Provision of food and monetary support by social services: for people with tuberculosis who were underweight or HIV-positive, despite tuberculosis not being included in the eligibility criteria;<br>• Community participation and churches: identified as important actors that provided informal social support;<br>• Definition of the role of the health and social assistance sector: in medical care and social assistance. | • Increased adherence rates to treatment, especially after financial support to reduce food insecurity |
| Boulanger et al, 2016 [14] | • Collaboration between the public and private sectors | • Some collaborations have demonstrated improvements in case detection, satisfactory treatment outcomes, and equitable access to treatment. |
| Weil et al, 2018 [15] | • Inclusion of the tuberculosis response in the Sustainable Development Goals Agenda: focus on universal access to care and prevention, sustainable financing, intensified research and multisectoral responsibility;<br>• "FIND. TREAT. ALL END TB" initiative: launched in 2018 by the WHO, seeks to ensure and expand universal health coverage, providing access to preventive and curative care for tuberculosis, in addition to intensifying the response to the disease and promoting technological innovations.<br>• Multisectoral responsibility framework: It was developed later with emphasis on intersectoral actions carried out in some countries [15]… | • The study did not present the results of the actions |
| World Health Organization, 2022 [16] | • In Brazil:<br>• Integration of multisectoral efforts and resources: in progress, in order to strengthen actions for the prevention, diagnosis, treatment and control of tuberculosis;<br>• Implementation of strategic plans: execution of cooperation agreements between ministries and organizations;<br>• Promoting civil society participation and creating review mechanisms: to monitor and evaluate activities related to combating tuberculosis; | • The integration of different sectors has achieved better health results<br>• In Portuguese-speaking countries, such as Angola, Mozambique and Timor-Leste, technical cooperation and surveillance activities have been carried out jointly with Brazil, aiming to strengthen the response to tuberculosis through international partnerships;<br>• Greater popular participation in tuberculosis control;<br>• Establishment of multisectoral coordination mechanisms at national and local levels: to ensure a comprehensive and integrated response to tuberculosis; |

*(Continued)*

 

**Table 3.** (Continued)

| Study | Intersectoral action approaches for tuberculosis control | Results of intersectoral actions for tuberculosis control |
|---|---|---|
| | • Involving different government sectors, such as health, education, justice, agriculture, environment, among others, to address the social and economic determinants that contribute to the tuberculosis endemic;<br>• Developing strategic plans: involving the participation of various stakeholders, such as academics, civil society, communities affected by tuberculosis and local program coordinators;<br>• Implementing work plans focused on tuberculosis control among the prison population: In prisons, including the introduction of molecular diagnostics in prison facilities and actions adapted to the needs of the prison population, this has been an effective approach.<br>• Community Committee for Monitoring Tuberculosis Research in Brazil: supporting community research initiatives to promote community participation in tuberculosis research;<br>• Promotion of social protection and combating stigma and discrimination related to the disease: through cooperation agreements between different ministries, such as Health, Social Development, Human Rights, among others;<br>• Collaboration between the Ministries of Health, Social Development and Human Rights: it has been fundamental to developing policies and programs that aim to guarantee access to social benefits, reduce the stigma associated with tuberculosis and promote inclusion and support for affected communities;<br>• Technical cooperation between countries;<br>• Exchange of knowledge and joint actions: in member countries of MERCOSUR and BRICS, the Ministry of Foreign Affairs of Brazil has promoted international cooperation activities for tuberculosis control, including: conducting research and studies on the co-infection of tuberculosis and COVID-19; | • During the COVID-19 pandemic, several countries and organizations have engaged in research to better understand the interaction between these two diseases and to develop effective prevention and treatment strategies for people affected by both conditions. |
| Rahevar et al, 2021 [17] | • Strengthening the link between health and education;<br>• Outbreaks provided opportunities for health education;<br>• Training in schools;<br>• Establishment of local policies and plans;<br>• The creation of national policies and local plans that coordinate the response to outbreaks is essential to ensure a harmonious and effective approach; | • The articulation between health and school: Provided opportunities for empowerment of students and their families with information about tuberculosis, reducing the stigma associated with the disease;<br>• In outbreak situations: Health education with health professionals, from the school and community about tuberculosis, increases the detection of tuberculosis cases, in addition to empowering those involved, resulting in greater adherence to treatment;<br>• Health education about tuberculosis in schools improves the ability of schools to detect potential cases of tuberculosis, which is crucial for a rapid and effective response;<br>• In situations of scarcity, it is important to consider mobilizing resources from neighboring areas or from external teams to ensure the continuity of health services. |
| World Health Organization, 2015 [18] | • Involvement of NGOs and Professional Associations:<br>• Participation of Public and Private Hospitals; | • Coordination and support, as in the case of the Myanmar Medical Association and the NGO Population Service International, was essential in assisting people with tuberculosis;<br>• Support in diagnosis, treatment and case management, as evidenced in Türkiye and Pakistan |

*(Continued)*

**Table 3.** (Continued)

| Study | Intersectoral action approaches for tuberculosis control | Results of intersectoral actions for tuberculosis control |
|---|---|---|
| | • Engagement of Private Laboratories: Diagnosis and referral of people with multidrug-resistant tuberculosis to health clinics, as exemplified in India and Vietnam;<br>• Support for people with tuberculosis: Actions such as providing social support, especially in cases of social vulnerability, as demonstrated in Türkiye and Pakistan. | • All approaches are important for greater control of the disease. |
| Spruijt et al, 2019 [19] | • Tuberculosis education and awareness for asylum seekers with tuberculosis.<br>• Collaboration with Partner Organizations<br>• Treatment Support | • Face-to-face education on tuberculosis and latent Mycobacterium tuberculosis infection, using professional interpreters, which improved adherence to screening and treatment;<br>• Partnerships between the health sector and agencies responsible for receiving and supporting asylum seekers increased adherence and facilitated the screening process;<br>• Personalized support from nurses, including the use of digital tools such as WhatsApp text messages, mobile alarms and delivery of weekly medication boxes, increased treatment completion rates. |
| Yuen et al, 2019 [20] | • Home-based care and monitoring of tuberculosis cases has expanded contact assessment and the use of preventive therapy. | • Increased participation in health assessments<br>• Provision of transportation vouchers and assistance in coordinating medical appointments facilitated access to health services for tuberculosis contacts in Carabayllo, Lima;<br>• The percentage of children under 5 years of age assessed increased from 70% to 90%.<br>• The intervention proved effective in improving tuberculosis management in the community. |
| Pescarini et al, 2017 [21] | • Conditional Cash Transfer Programs – Bolsa Família in Brazil<br>• Screening strategies for active and latent tuberculosis among migrants: to reduce the burden of disease and the risk of transmission in Latin American countries, e.g., Brazil, Argentina, Bolivia, Chile and Paraguay; _ Free access to tuberculosis diagnosis and treatment for migrants: regardless of their migratory status in Brazil | • Implementation of programs such as "Bolsa Família" in Brazil, which have shown a positive impact on nutrition, food security, use of health services and reduction of maternal and infant mortality;<br>• The universal right to health in Brazil facilitates access to health actions and services. |
| Sharma et al, 2019 [22] | • Attention to vulnerable groups:<br>• Provision of health care in the truck drivers' workplace, directly at the transport hubs:<br>• Coordination between the health and work sectors<br>• Use of infrastructure and human resources: existing in the Sexually Transmitted Infections (Khushi) clinics and Testing and Counseling Centers to provide tuberculosis diagnosis and treatment services in an accessible manner in the work sector;<br>• Implementation of interpersonal sessions, transporters' meetings, group sessions and health education, the basic principles of tuberculosis, prevention and treatment measures to raise awareness of the problem. | • Assistance to truck drivers with personalized actions was essential for greater control of the disease;<br>• Cooperation with the labor sector in transport centers: facilitated access to health (TB testing and treatment) for truck drivers in India, considering working conditions;<br>• During the study, 14,644 truck drivers participated in the sessions, to increase knowledge about tuberculosis and the importance of early diagnosis;<br>• 297 truck drivers were scheduled for tuberculosis tests, of which 283 actually underwent the tests, resulting in 10 positive diagnoses. |

Source: Research Data, Natal–RN, 2024.

comprehensive approach. The strategy advocates universal access to diagnosis, treatment and prevention, with sustainable financing and the implementation of technological innovations [1].

However, in the prison context, the challenge is intensified by the difficulty of ensuring equitable and continuous access to these innovations, in addition to the structural barriers that limit the scope of interventions. Thus, although local intersectoral initiatives have shown positive results, it is evident that a global and integrated strategy, such as the End TB Strategy, needs to be adapted to contexts such as prisons, where conditions of overcrowding, scarcity of resources and stigma require more specific and coordinated actions between the health and justice sectors. Only in this way will it be possible to achieve global goals and effectively control tuberculosis in such vulnerable environments and in the community.

Studies show that the increase in incarceration rates in many countries has intensified the impact of tuberculosis among populations deprived of their liberty, with consequences that transcend prison walls and compromise the control of the disease within communities [27,28]. The overload of prison systems and the precarious health conditions in these institutions aggravate the scenario, reinforcing the need for intersectoral action.

Countries such as Austria, Spain and the United Kingdom stand out for the use of screening and referral of people deprived of liberty with multidrug-resistant tuberculosis (MDR-TB) to specialized health services, a practice that has increased detection and facilitated treatment. In Bulgaria, the implementation of clinics within prisons for the treatment of tuberculosis also brought significant results [12].

In Brazil, the introduction of molecular diagnostics in prisons has increased the effectiveness of these initiatives, promoting early detection and appropriate treatment of cases [15], although more intense efforts are still needed. These findings underscore the urgency of a continued commitment between the ministries of health and justice in each nationality to ensure early detection and treatment since the entry of persons deprived of their liberty into the prison system [27,28].

Another aspect of great relevance is social support, evidenced as a central component in the fight against tuberculosis, especially among vulnerable populations. The provision of food and financial support, as observed in Mozambique and other affected regions, has been shown to be effective in improving treatment adherence and reducing therapeutic abandonment [14].

Boccia, Pedrazoli and Wingfield [29] argue that social protection is fundamental to prevent poverty and reduce socioeconomic vulnerability, increasing the ability of individuals to protect themselves economically and thus complete tuberculosis treatment.

This support is essential not only to avoid treatment interruption, but also to mitigate the costs associated with disease management, as observed in several of the interventions analyzed. These initiatives are particularly relevant in populations with difficult access, such as migrants, truck drivers and individuals deprived of liberty, where living conditions aggravate barriers to treatment, reinforcing the importance of policies that integrate social protection and health [12,16,21,22].

The relevance of social protection programs in tuberculosis control is largely corroborated by several studies that highlight their ability to improve treatment adherence and clinical outcomes, especially in vulnerable populations [13,20,21]. Social protection policies, as evidenced in some studies, play a central role in the intersectoral response to tuberculosis control [30].

Likewise, the integration of income transfer programs, such as Bolsa Família in Brazil, has proven to be an effective strategy in improving the living and health conditions of people affected by tuberculosis, encouraging treatment adherence and reducing abandonment in socioeconomic vulnerability, promoting advances in tuberculosis control [21].

A study in Brazil revealed that the cure rate was 7.6% higher among people with tuberculosis who received the benefit, in addition to recording a 7% reduction in treatment abandonment, evidencing the positive impact of financial support on treatment adherence [31]. In addition, the provision of transportation vouchers and assistance in coordinating medical appointments, as occurred in Carabayllo, Lima, substantially increased the rates of testing and treatment of people with latent infection [20].

Another example of success was observed in Turkiye and Pakistan, where social support, especially in situations of vulnerability, proved essential for adherence to treatment [18]. These data reinforce the argument that the high cost associated with medical care for tuberculosis, such as transportation, medications and loss of income, perpetuates a cycle of poverty and disease, as highlighted by Ernst [32]. Families facing financial difficulties tend to postpone seeking care, which can result in late diagnoses, worsening of the clinical picture and greater spread of the disease.

Thus, the economic burden of tuberculosis affects both the individuals affected and the health systems and society, due to the high costs of treatment and the social consequences associated with the disease. To mitigate these challenges, public policies that address economic barriers are essential to ensure universal access to timely and adequate care [33].

In this sense, free access to diagnosis and treatment for migrant populations is an essential measure to maximize adherence and control the disease in these communities and territories [34]. This approach is shared by countries such as Brazil, Argentina, Chile, Paraguay and Bolivia [16,21].

In addition to access to treatment, the participation of NGOs and the community has been a key component in tuberculosis control strategies. In Spain, NGOs were decisive in the care of people with MDR-TB, offering support in supervised treatment, housing and employment for migrants, which facilitated the continuity of care and contributed to therapeutic success [12]. Similar initiatives in Myanmar, through coordination between professional associations and NGOs, such as the Myanmar Medical Association and Population Service International, have reinforced the expansion of access and quality of services related to tuberculosis [18].

In Brazil, a qualitative study conducted in Campina Grande-PB revealed that social support was decisive for the success of the treatment. People with tuberculosis who maintained healthy relationships with family and friends felt more encouraged to complete the treatment, highlighting the positive role of the support network in the therapeutic process. However, when this network was fragile or non-existent, the stigma surrounding the disease made adherence difficult, with many individuals choosing to hide the diagnosis, which represented a significant barrier to appropriate treatment [35].

To address these obstacles, community initiatives have been shown to be effective in raising awareness about tuberculosis, especially among hard-to-reach groups such as people in social vulnerability, migrants, and individuals deprived of liberty. The involvement of churches and other community actors was essential to provide informal social support, promote treatment adherence, and reduce the stigma associated with the disease [13]. In contrast, the lack of awareness about tuberculosis among those affected and their support networks has been a factor that delays access to diagnosis and treatment, as observed in a study [36].

Thus, the findings reinforce the importance of an intersectoral and integrated approach, which not only ensures universal and free access to tuberculosis diagnosis and treatment, but also includes social support as a strategic tool to address the economic and social barriers that affect treatment adherence, especially in vulnerable populations.

It was also revealed that education and the articulation between the health and education sectors have been fundamental in training students and families, promoting awareness of tuberculosis and strengthening both prevention and early detection of the disease.

In the Eastern Pacific region (China, Japan, Republic of Korea and Mongolia), as observed in the Study [17], this integration contributed significantly to the empowerment of those involved, which resulted in greater adherence to treatment and in the reduction of stigma associated with the disease. Similarly, a prospective study conducted in China revealed that schools, as environments conducive to the spread of tuberculosis, also play a crucial role in identifying cases of active tuberculosis and latent infection [37].

In addition to the role of schools, intersectoral collaboration, which involves the participation of sectors such as health, justice, education and non-governmental organizations, is essential for the success of tuberculosis control actions. In Brazil, the Community Committee for Monitoring Tuberculosis Research has shown that civil society engagement contributes to the development of more inclusive strategies, adapted to the needs of communities [16].

It is evident that intersectoral actions, when combined with public policies focused on equity and social inclusion, provide a more robust response to the fight against tuberculosis. The engagement of civil society, community organizations and NGOs, as seen in Spain and Myanmar [12,18], strengthens support networks for people with tuberculosis and ensures that the treatment approach is not limited to the clinical scope, but also takes into account the social and economic needs of the affected populations.

This is because promoting inclusive public health, involving education, awareness and combating stigma, has been equally essential. The articulation between the education sector and health, as seen in the Eastern Pacific Region and China, has proven effective in disseminating information about tuberculosis, promoting early detection and reducing stigma that negatively affects treatment adherence [17,19].

Thus, Health education, both in the school environment and in the community, is also a central tool in the fight against tuberculosis. Educational initiatives aimed at raising awareness, such as those implemented in the Netherlands for multilingual populations [19], are fundamental to demystify the disease and reduce stigma, promoting a supportive environment that favors treatment adherence. This type of approach is especially important in marginalized populations, such as migrants, truck drivers and people deprived of liberty, since they face additional barriers to accessing health care [13,22].

The promotion of health education campaigns, with a focus on reducing stigma and improving treatment adherence, is fundamental to the success of global tuberculosis control strategies [38]. In addition, the articulation between the various sectors, when accompanied by consistent and sustainable public policies, offers a promising way to reduce the incidence of tuberculosis and improve the living conditions of affected populations.

Collaboration between the public and private sectors has been shown to be an effective approach in improving outcomes related to tuberculosis control, especially in countries with weakened health systems. Public-private partnerships have been essential in improving case detection, satisfactory treatment outcomes, and equitable access to health services, as observed in Turkiye and Pakistan [18].

Similarly, in India and Vietnam, the involvement of private laboratories contributed significantly to the early detection and appropriate referral of people with multidrug-resistant tuberculosis, strengthening the integration of resources and competence from different sectors [18]. These findings are corroborated by studies carried out in Bangladesh, which highlight that, in nations with insufficient health coverage, the articulation between the public and private sector is crucial to ensure accessibility and effectiveness in the treatment of tuberculosis [39].

In addition, the impaired financial situation associated with the scarcity of public health services or affordable value can lead symptomatic individuals to seek help from informal professionals, such as healers, who do not have the necessary knowledge for the proper management of the disease, resulting in inaccurate diagnoses and delay in starting treatment [39].

In addition to these partnerships, the implementation of intersectoral public policies has played a key role in effectively tackling tuberculosis. The inclusion of the disease response in the agenda of the Sustainable Development Goals (SDGs) has emphasized the need for integrated approaches, involving sustainable financing, intensified research and multisectoral responsibility. The initiative "FIND. TREAT. ALL END TB", launched by the World Health Organization (WHO), promotes global coordination of efforts, bringing together diverse sectors and organizations to intensify the response to tuberculosis and facilitate the development of innovative strategies [15,16].

In Brazil, integration between the ministries of health, education, justice, agriculture and the environment has been a promising practice to address the social and economic determinants of tuberculosis, reflecting the importance of effective management of human and material resources [16]. This articulation is also observed in Portuguese-speaking countries, such as Angola, Mozambique and Timor-Leste, where technical cooperation with Brazil has strengthened surveillance and strategies adapted to local realities. International collaboration, through initiatives between Mercosur and BRICS members, also facilitates the exchange of knowledge and the execution of joint actions for tuberculosis control [16].

During the COVID-19 pandemic, the interaction between tuberculosis and coronavirus became a research focus for several countries, revealing the need for integrated prevention and treatment strategies. International cooperation, highlighted by WHO, highlights the importance of intersectoral collaboration to boost the development of new strategies and the advancement of research related to tuberculosis [40]. The implementation of collaborative research networks, such as the Community Committee for Tuberculosis Research [16], has been an important measure to foster the exchange of knowledge and technologies aimed at combating the disease.

Thus, the continuity of intersectoral articulation and international cooperation is important for the development of sustainable and effective public policies to control tuberculosis. Collaboration between countries, as observed between Mercosur and BRICS members and the strengthening of research networks are examples of strategies that have the potential to generate positive impacts on both the diagnosis and treatment of the disease [16]. This collective effort aims not only to expand access to treatment, but also to develop technologies and innovations that can facilitate early detection and improve treatment adherence, especially in vulnerable populations.

The results of this review emphasize the relevance of the articulation between multiple sectors, such as health, education, social assistance, justice and work, showing that multi sectoral coordination is fundamental for the success of tuberculosis control policies. The continuous mobilization of resources and the implementation of technological innovations, combined with the strengthening of the role of the community, are fundamental to overcome the social and economic barriers that hinder access to treatment and therapeutic adherence, especially for vulnerable people.

## Conclusion

The evidence indicates that the most used approaches involved educational campaigns focused on raising awareness and empowering the individual and the collectivity, as well as providing social support actions, such as income transfer programs and support from NGOs and communities. The strengthening of the coordination among health services, schools, social assistance, and prisons, as well as the involvement of public-private partnerships and international cooperation, were also highlighted.

These actions aimed to increase treatment adherence, reduce stigma, improve access to health services, facilitate early diagnosis, and provide financial and food support, especially among vulnerable populations, such as those with socioeconomic vulnerability, people with HIV, prisoners, truck drivers, and migrants. The most significant results showed improvements in cure rates, a reduction in disease transmission and an increase in the social empowerment of those affected and their families, highlighting the positive impact of intersectoral approaches.

Among the evaluated approaches, those that combined health education actions, social support, technology use – such as text messages and alarms to improve treatment adherence, when carrying out treatment supervision – and direct intervention in the community showed the best results. These initiatives promoted greater user commitment, reduced treatment abandonment and facilitated the diagnosis of latent cases, especially in vulnerable populations. International cooperation and public-private partnerships have also proven to be essential to ensure the sustainability and expansion of interventions.

The limitations of this survey include the possibility of excluding relevant studies, particularly when intersectoral collaboration was not clearly described in the abstracts. In addition, studies addressing intersectoral actions in tuberculosis control without the explicit involvement of the health sector may not have been captured by the adopted search strategies, which could limit the understanding of the complete picture of these approaches and outcomes.

Furthermore, the scarcity of studies from countries with a high burden of tuberculosis in this study, possibly due to language restrictions or limitations in accessing databases, undermines the generalization of the results. Therefore, the need for future and more in-depth research on intersectoral actions in terms of tuberculosis control should be reinforced,

especially in contexts of high disease incidence, in order to expand knowledge and improve strategies for confronting tuberculosis globally.

Finally, the results reinforce that coordinated approaches, adapted to the local context and directed towards social inclusion and protection, have a greater potential to enhance progress in tuberculosis control. These lessons underscore the importance of strengthening intersectorality and social and technological innovation in future public policies, which are essential for achieving the global goals of the End TB Strategy.

## Supporting information

**S1 Search Strategies.** The document details the process of electronically capturing studies into databases.
(DOCX)

**S2 File. Spreadsheets for data storage.** This spreadsheet compiles the data collected for this stud.
(XLSX)

**S3 Fig. Checklist Prisma-ScR.** Of the study selection process.
(TIF)

**S4 Fig. Conceptual structure of approaches and results of intersectoral actions in tuberculosis control, Natal RN, 2024.** The figure presents the visual synthesis of the approaches and results identified in the scoping review. It structures and systematizes the information to improve the understanding of the mapped intersectoral approaches and results.
(TIF)

## Acknowledgments

Special thanks to Mônica Karina Santos Reis, librarian of the Sector Library Prof. Alberto Moreira Campos of the Department of Dentistry of the Federal University of Rio Grande do Norte for his assistance in guiding the search strategies for this scoping review.

## Author contributions

**Conceptualization:** Rosiane Davina da Silva, José Mateus Bezerra da Graça, Eslia Maria Nunes Pinheiro, Elisângela Franco de Oliveira Cavalcante.

**Data curation:** Rosiane Davina da Silva, Erica Rayane Galvão de Farias.

**Formal analysis:** Rosiane Davina da Silva, Elisângela Franco de Oliveira Cavalcante.

**Funding acquisition:** Elisângela Franco de Oliveira Cavalcante.

**Investigation:** Rosiane Davina da Silva, Erica Rayane Galvão de Farias.

**Methodology:** Rosiane Davina da Silva, Elisângela Franco de Oliveira Cavalcante.

**Project administration:** Rosiane Davina da Silva, Elisângela Franco de Oliveira Cavalcante.

**Resources:** Rosiane Davina da Silva.

**Supervision:** Rosiane Davina da Silva, Elisângela Franco de Oliveira Cavalcante.

**Visualization:** Rosiane Davina da Silva, Elisângela Franco de Oliveira Cavalcante.

**Writing – original draft:** Rosiane Davina da Silva.

**Writing – review & editing:** Rosiane Davina da Silva, Erica Rayane Galvão de Farias, José Mateus Bezerra da Graça, Eslia Maria Nunes Pinheiro, Elisângela Franco de Oliveira Cavalcante.

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
