## [Decision Letter · Decision Letter 0]

Dear Dr. Silva,

Thank you for submitting your manuscript to PLOS ONE. After careful consideration, we feel that it has merit but does not fully meet PLOS ONE’s publication criteria as it currently stands. Therefore, we invite you to submit a revised version of the manuscript that addresses the points raised during the review process.

We look forward to receiving your revised manuscript.

Kind regards,

Zinia Thajudeen Nujum

Academic Editor

PLOS ONE

“This study was financed in part by the Coordenação de Aperfeiçoamento de Pessoal de Nível Superior - Brasil (CAPES) - Finance Code 001.”

Additional Editor Comments:

ACADEMIC EDITOR Comments :

Approaches and results of intersectoral actions for tuberculosis control in the world: A scoping review

General

The area of work is relevant  and good attempt

Specific comments

The definition of intersectorality has not been provided , the PCC is not framed in a way to capture the question/objective and it appears that therefore the relevant studies have not been picked up

Is population – Tuberculosis ?

I think the concept needed to be broken down to name the sectors that we are specifically looking for and defined more accurately with search terms to represent all these.

Why did the authors choose to register in Open Science Framework which is a broader platform and not PROSPERO  meant specifically for reviews

Line 128 – typo – intersectorality

Context (C):  National Tuberculosis Control Programs, National Tuberculosis Control Policy

130 and Tuberculosis Control Actions, Public Health  -  the national control program /policy , actions , public health – none of these figure in the search  - Did the study really look at studies with intersectoral collaboration within the framework of national programs ?

Methods – Was there a limit year ? Were studies in all languages included ?

Include the search strategy and results from at least three databases in the supplementary material

Table 3 – Yuen et al study the percentage of children under 5 years of age evaluated……(?increased ) from 70% to 90% was found. Was found may not be needed – check the sentence

Table 3 – the approaches and results may be split into separate columns

Conclusion – What is concluded from the paper are things that are already known. Can you include some new information from the article – since the objective is approaches and results of intersectoral actions can you bring in some best practice/approach in intersectoral action with remarkable results in TB control.

Reviewers' comments:

Reviewer's Responses to Questions

**Comments to the Author**

1. Is the manuscript technically sound, and do the data support the conclusions?

Reviewer #1: Partly

Reviewer #2: Partly

2. Has the statistical analysis been performed appropriately and rigorously?

Reviewer #1: N/A

Reviewer #2: No

3. Have the authors made all data underlying the findings in their manuscript fully available?

Reviewer #1: Yes

Reviewer #2: Yes

4. Is the manuscript presented in an intelligible fashion and written in standard English?

Reviewer #1: Yes

Reviewer #2: Yes

Reviewer #1: 1. The review addresses a relevant topic, and the scoping review follows a strong methodology. However, the result section may be modified for better readability of the article.

2. What made the decision to include reviews, manuals, technical and governmental publications to be included in the review and editorials, experience reports, annals and books were excluded. The distinction is very narrow between many of the included and excluded categories, which can bring selection bias. Moreover, as reviews can use editorials and books along with original article, what is the point for exclusion criteria?

3. For the last column of table 2, the term ‘sample’ may be more appropriate than the term ‘population’.

4. Pushing data to the Table 3 is making both the table and the article in comprehensible. Better give the approaches and major intersectoral actions as short phrases or bullet points in the tables and give the details as running text.

5. The authors give two figures figure 2 and 3 illustrating approaches and results of intersectoral actions. However, the data is highly fragmented and listed under the candidate studies. It would be great if the authors can give a conceptual framework or a diagram illustrating the intersectoral actions in global TB control based on the collective wisdom they obtained from the review. The figure 2 and 3 could be shifted to the appendix if such a framework/diagram is made available.

6. The discussion and conclusion part of the manuscript should be significantly trimmed to get a focused view.

Reviewer #2: The paper addresses an important topic on intersectoral actions on TB. The aim was to map various approaches and results of intersectoral actions for TB and the authors tried to do that through a scoping review.

However, the reviewer could feel many significant issues

1. The search strategy and approach did not seem to have got all good documents and also did not have a good inclusion and exclusion criteria. Word intersectoral is too broad and it is difficult to get the correct studies/documents. It would have been good to relook, if possible and include the relevant studies. There is numerous adaptation of Multi sectoral accountability framework of WHO in various countries/regions with documented case studies, all those seems to have missed.

2. The included studies also seem to be very narrow and not adequately reflecting the theme. Some of them are just commentaries, many were reviews, some are very focused on outbreaks context, some on private sector engagement, some on community led monitoring. If such studies are included, there are a lot of similar studies (private sector engagement and community engagement itself have many good studies across the globe).

Request to revisit and look for important studies.

3. Results of intersectoral actions are not very clear from the review, may need to synthesize better

4. Conclusion is not at all backed up with the results. Conclusion should be based on the results.

Minor issues

Word eradication, elimination, control is being used interchangeably without understanding the terminologies. Please relook and try to use appropriate terminologies.

End TB and TB Elimination are used interchangeably- please relook (eg: Line 87-89- the statement seems wrong; it should be the other way. Also SDG is never intending to eliminate TB)

**Do you want your identity to be public for this peer review?** For information about this choice, including consent withdrawal, please see our Privacy Policy

Reviewer #1: **Yes: ** Gayathri AV

Reviewer #2: **Yes: ** RAKESH PS

---

## [Author Response · Author response to Decision Letter 1]

9 Mar 2025

Dear Editor, we are very grateful for your valuable contributions, which were accepted. They were very important to make the manuscript more robust and suitable for the journal. The response will be made according to the notes, and if you find any other details that need improvement, please let us know, and we will be happy to improve them.

1. A review was carried out on the adequacy of the manuscript to the PLOS ONE style according to the sources indicated, as noted in the file.

2. The funding source was corrected in the cover letter and in the additional information section regarding financial disclosure.

3. The support files used for this review are all attached to the submission to the journal, as noted in the attachments Support file - Spreadsheets for storing data; Support Files - Search Strategies and PRISMA-ScR-Fillable Checklist and we are sending a data sharing plan to the journal signed by the authors.

4. The legends for the supplementary information files were included at the end of the Lines manuscript 489 to 499 and all citations were reviewed to properly match the PLOS ONE style

- Response to additional comments from the Academic Editor

1. The definition of Intersectorality was inserted: Lines: 69-74.

2. Is population tuberculosis? No, according to JBI recommendations, the mnemonic combination of PCC is used, which can correspond to Problem or population, Concept and Context. Therefore, the term “Problem” was used and not “Population” in the respective study; however, if a replacement for the term “Population” is necessary, we can do so.

3. Regarding the concept, in this study we proposed to identify approaches and results of intersectoral actions in the studies that were found. Therefore, we did not limit ourselves to a precise search for the sectors, although it was clear that the sectors of social assistance, community and family support, collaborations between countries, public-private relations, schools and the prison system are acting in an intersectoral manner in tuberculosis care. According to this specification, if we break down the concept by pointing out the possible sectors, more elements may appear, but not necessarily in a collaborative manner, which would be at odds with the primary objective of this study.

4. The choice of OSF was made due to its broader scope and its frequent use by researchers in Brazil and abroad, including for review purposes. Although we are not very familiar with the Prospero platform, in future studies we propose to review this issue and get to know it more deeply. We appreciate the suggestion.

5.The translation mistake in line 128 regarding intersectorality was corrected: Line: 108.

6. Although it was not mentioned due to the scarcity of studies on the programs, tuberculosis control in all countries is directly managed by National Tuberculosis Control Programs, which follow a specific hierarchy as recommended by The End-TB strategy. These programs are decentralized, and the World Health Organization (WHO) has assigned National Tuberculosis Control Programs the responsibility of implementing tuberculosis control actions in the countries. Although some studies do not directly mention the programs, they indicate that the actions are coordinated by local health. For this reason, the study included these studies that portray intersectoral action within these structures with a focus on their approaches and results; we did not intend to analyze or evaluate such actions in each nationality.

7. The year threshold is shown in the line: 162.

8. Studies in all languages were not included, as stated in the Lines: 159-160.

9. All search data from the supplementary file “Search strategies” are available according to their adaptations and number of studies found 488-500.

10. Correction made in Table 3 in the study by “Yuen et al”, the term “were found” was removed

11. Table 3 was reformulated with the creation of the results column, to separate the approaches from the results, as requested: Line: 207.

12. Adjustments in the conclusion were made: Lines: 456-486.

Reply to Reviewer 1.

1. Adjustments in the results section were made, Lines: 207-216.

2. Regarding the annals and editorials, they are short and superficial texts. Regarding the books, it would not be possible to include updated books on the subject. The decision to include reviews, manuals, technical and government publications in the review, while editorials, experience reports, annals and books were excluded, is based on the need to ensure the quality, reliability and applicability of the data analyzed. Systematic reviews, manuals and institutional publications generally follow strict methodological criteria and are supported by scientific evidence or normative guidelines, which contributes to greater robustness in the findings. It is recognized that some excluded categories could contain relevant information, but it was decided to focus on sources that present a higher degree of methodological standardization and a lower degree of subjectivity. The exclusion of editorials and books is due to the fact that these materials often do not follow the same peer review criteria and may contain more opinionated interpretations. Regarding the possibility of reviews including editorials and books in their analyses, it is worth noting that the selection of reviews took into account their search and inclusion methodology, in order to minimize bias in the interpretation of the results. Thus, the exclusion criteria were established to improve the consistency and validity of the review performed.

3. The term “sample” was replaced in Table 2. Line: 195.

4. Adjustment was made; the results remain in the table, but the conceptual structure (Fig 3) presents the data with short sentences: Lines: 205-216.

5. The conceptual structure was created Fig 2, Line: 212.

6. The discussion and conclusion were reduced as suggested: Lines: 218-486, if it is necessary to reduce it even further, please do not hesitate to let us know. We did not reduce it further because we were concerned about leaving out important results.

-Reply to Reviewer 2

1. We appreciate your comments on the study. The process of retrieving studies was carried out in a detailed manner, aiming to achieve the best results that met the objective of the study, as verified in the support file of the search strategies. As an example, in the gray literature, especially in the “WHO Global Research” database, we carefully examined 133 publications. When these publications mentioned multisectoral/intersectoral actions in some location, we always consulted the original sources. However, we found that most presented more intersectoral strategies than concrete actions effectively carried out (approaches and their results), which is at odds with our analysis objective. For this reason, these publications were not included in our analysis. It is important to emphasize that the objective was to evaluate the tuberculosis control actions that were carried out and their results; the case studies found reflected more strategies/recommendations.

2. It is important to clarify that the primary intention of our study was to focus on intersectoral collaboration for tuberculosis control. The identified sectors emerged throughout the research process and were not previously anticipated. Even if the sectors were included, the results could be expanded, but not necessarily in a collaborative manner, which would be at odds with the primary objective of this study.

3. Changes were made in Table 3 to separate the results, Line: 207.

4. Changes were made in the conclusion, lines: 456-486.

5. Responses to minor issues:

5.a. Revised use of terminology and translation

5.b. Corrected, Lines: 61-63.

Note: We are resending the PRISMA checklist and supporting files S1 Search Strategies, S2 Spreadsheets for storing data, with the term “PubMed” replaced by “Medline”, as Medline is the more precise term when referring to the database.

---

## [Decision Letter · Decision Letter 1]

Dear Dr. Silva,

Thank you for submitting your manuscript to PLOS ONE. After careful consideration, we feel that it has merit but does not fully meet PLOS ONE’s publication criteria as it currently stands. Therefore, we invite you to submit a revised version of the manuscript that addresses the points raised during the review process.

We look forward to receiving your revised manuscript.

Kind regards,

Zinia Thajudeen Nujum, M.D

Academic Editor

PLOS ONE

Additional Editor Comments:

The comments on the revised manuscript from reviewer is given below. Please work on them for reconsidering publishing your article. 

Thanks for the reply and addressing the comments.

The reviewer still feel that conclusion is not in alignment with the objectives and results. There are many generic statements which are not based on the results of this study. Based on the objective, a reader would like to hear as conclusion: What were the approaches used, what were the results, which ones were better and what are the learnings.

Inclusion and exclusion of the studies are not clear yet to the reviewer. There are many good studies/reports excluded for the review. Local Government stewardship/ Private sector engagements/Community engagements etc. are some examples where lots of studies are available from high burden countries. There are not much studies from the high burden countries included in the scoping review. Request to highlight those as limitations.

Word elimination, control is being used interchangeably. Request to relook again. Authors generally used TB control, still Elimination appears at 9 places. Kindly relook and choose appropriately. Also SDG never intends to 'eliminate' TB. Request to relook again (line 61-63)

Also there is scope for making the writing crisp and comprehensive.

Reviewers' comments:

Reviewer's Responses to Questions

**Comments to the Author**

Reviewer #1: All comments have been addressed

Reviewer #2: (No Response)

2. Is the manuscript technically sound, and do the data support the conclusions?

Reviewer #1: Yes

Reviewer #2: Partly

3. Has the statistical analysis been performed appropriately and rigorously?

Reviewer #1: N/A

Reviewer #2: Yes

4. Have the authors made all data underlying the findings in their manuscript fully available?

Reviewer #1: Yes

Reviewer #2: Yes

5. Is the manuscript presented in an intelligible fashion and written in standard English?

Reviewer #1: Yes

Reviewer #2: Yes

Reviewer #1: (No Response)

Reviewer #2: Thanks for the reply and addressing the comments.

The reviewer still feel that conclusion is not in alignment with the objectives and results. There are many generic statements which are not based on the results of this study. Based on the objective, a reader would like to hear as conclusion: What were the approaches used, what were the results, which ones were better and what are the learnings.

Inclusion and exclusion of the studies are not clear yet to the reviewer. There are many good studies/reports excluded for the review. Local Government stewardship/ Private sector engagements/Community engagements etc. are some examples where lots of studies are available from high burden countries. There are not much studies from the high burden countries included in the scoping review. Request to highlight those as limitations.

Word elimination, control is being used interchangeably. Request to relook again. Authors generally used TB control, still Elimination appears at 9 places. Kindly relook and choose appropriately. Also SDG never intends to 'eliminate' TB. Request to relook again (line 61-63)

Also there is scope for making the writing crisp and comprehensive.

**Do you want your identity to be public for this peer review?** For information about this choice, including consent withdrawal, please see our Privacy Policy

Reviewer #1: **Yes: ** Gayathri A V

Reviewer #2: **Yes: ** RAKESH PS

---

## [Author Response · Author response to Decision Letter 2]

21 May 2025

Dear Publisher,

Reply to Reviewer 2.

We would like to thank you very much for your valuable contributions, all of which have been accepted. They were of great importance in making the manuscript more robust and suitable for the journal. The reply will be drafted according to the observations made. If there are any other details that need improvement, please let us know; we will be happy to improve them.

1. Redrafted the “Final Considerations”. Lines: 452-490.

2. Adjustments made to the study’s limitations. Lines: 474-485.

3. Adjustments made to the terms “Eliminate” and “Elimination”. Lines: 19; 58; 63; 65; 256. All excluded.

Thank you!

---

## [Decision Letter · Decision Letter 2]

Approaches and results of intersectoral actions for tuberculosis control in the world: A scoping review

PONE-D-24-48617R2

Dear Dr. Rosiane Davina da Silva,

We’re pleased to inform you that your manuscript has been judged scientifically suitable for publication and will be formally accepted for publication once it meets all outstanding technical requirements.

Kind regards,

Zinia Thajudeen Nujum, M.D

Academic Editor

PLOS ONE

Reviewers' comments:

Reviewer's Responses to Questions

**Comments to the Author**

Reviewer #2: All comments have been addressed

2. Is the manuscript technically sound, and do the data support the conclusions?

Reviewer #2: Partly

3. Has the statistical analysis been performed appropriately and rigorously?

Reviewer #2: N/A

4. Have the authors made all data underlying the findings in their manuscript fully available?

Reviewer #2: Yes

5. Is the manuscript presented in an intelligible fashion and written in standard English?

Reviewer #2: Yes

Reviewer #2: (No Response)

**Do you want your identity to be public for this peer review?** For information about this choice, including consent withdrawal, please see our Privacy Policy

Reviewer #2: **Yes: ** RAKESH PS

---

## [Editor Report · Acceptance letter]

PONE-D-24-48617R2

PLOS ONE

Dear Dr. Silva,

I'm pleased to inform you that your manuscript has been deemed suitable for publication in PLOS ONE. Congratulations! Your manuscript is now being handed over to our production team.

Kind regards,

on behalf of

Dr. Zinia Thajudeen Nujum

Academic Editor

PLOS ONE